# Effects of Cavitation Jet Treatment on the Structure and Emulsification Properties of Oxidized Soy Protein Isolate

**DOI:** 10.3390/foods10010002

**Published:** 2020-12-22

**Authors:** Mingyu He, Changling Wu, Lijia Li, Li Zheng, Tian Tian, Lianzhou Jiang, Yang Li, Fei Teng

**Affiliations:** 1College of Food Science, Northeast Agricultural University, Harbin 150030, China; hemingyu11301@163.com (M.H.); wuchangling0805@163.com (C.W.); lilijia19950820@163.com (L.L.); 14794306051@163.com (L.Z.); tiantian@neau.edu.cn (T.T.); jlz0109@neau.edu.cn (L.J.); 2Harbin Institute of Food Industry, Harbin 150030, China; 3Heilongjiang Academy of Green Food Science, Harbin 150030, China

**Keywords:** cavitation jet, emulsifying properties, peroxyl radical, soy protein isolate, structure

## Abstract

This study examined the ability of cavitation jet processing to regulate the oxidation concentrations with 2,2’-azobis (2-amidinopropane) dihydrochloride (AAPH) (0.2, 1, and 5 mmol/L) and the structure and emulsification of soy protein isolate (SPI). The tested properties included particle size distribution, hydrophobic properties (sulfhydryl group (SH) and disulfide bond (S-S) contents, surface hydrophobicity (*H*_0_)), emulsifying properties (particle size and ζ-potential of emulsions, emulsification activity index (EAI), and emulsification stability index (ESI)), as well as conformational characteristics. The high shear force of cavitation jet treatment reduced the particle size of oxidized SPI and distributed uniformly. Cavitation jet (90 MPa)-treated SPI (AAPH with 1 mmol/L) demonstrated a high *H*_0_ (4688.70 ± 84.60), high EAI (71.78 ± 1.52 m^2^/g), and high ESI (86.73 ± 0.97%). The ordered secondary structure (α-helix and β-turn content) of SPI was enhanced by the cavitation jet. Meanwhile, the distribution of SPI-oxidized aggregates was observed under an atomic force microscope. Therefore, cavitation jet processing combined with oxidation treatment is an effective method to improve the characteristics of SPI and has potential industrial application prospects.

## 1. Introduction

Soy protein is a high-quality plant-derived protein. Due to its unique nutritional value and excellent functional properties, soy protein has become one of the most important food ingredients widely used in the food industry [1,2]. At present, soybean protein isolate (SPI) is prone to oxidation and aggregation during storage and transportation, resulting in the loss of protein nutrition, quality degradation, and loss of some functional properties, such as solubility, gelation, and emulsification [3,4]. Protein oxidation is the covalent modification of a protein directly induced by reactive oxygen species or indirectly induced by reaction with secondary by-products of oxidative stress [5]. Oxidation can partially unfold proteins as follows: (i) the exposed groups are further oxidatively modified and (ii) oxidized aggregates are formed through hydrophobic interactions and electrostatic interactions [6,7]. Therefore, researchers are interested in developing simple and sustainable processing methods to effectively regulate the oxidative aggregation behavior of soybean proteins and inhibit the degradation of protein products caused by oxidative damage.

Traditionally, the method of plant protein oxidation is induced mainly by oxidants (i.e., 13-hydroperoxyoctadecadienoic acid, malondialdehyde, acrolein, and 2,2′-azobis (2-amiylpropane) dihydrochloride (AAPH)) [8]. A water-soluble free radical initiator, AAPH is preferred to generate peroxy radicals at a constant rate [9,10]. Thermal decomposition of AAPH leads to the formation of carbon-centered free radicals, generating peroxy radicals under aerobic conditions [11,12,13]. In recent years, researchers have used AAPH-derived peroxy radicals to induce SPI oxidative modification and found that these radicals attack almost all types of amino acid side chains. AAPH oxidation leads to the formation of carbonyl groups, the degradation of free sulfhydryl groups, and the formation of dihydrotyrosine in SPI. Oxidative modification results in the breaking of the main chain of proteins, a decrease in solubility and interfacial activity, and the formation of oxidative aggregates [3,14].

At present, improving or inhibiting the adverse effects of the excessive oxidation of proteins is an urgent issue. Many physical treatments, such as high-pressure homogenization, ultrasound, and high hydrostatic pressure, have been widely used to improve the physical and chemical properties of soy protein [15,16,17,18]. Previous studies have emphasized the potential of using new processing technologies to change the structure and functional performance of SPI. Cavitation jet technology is a new type of food processing technology, which utilizes high pressure and cavitation combined with high-speed impact, high-frequency vibration, instantaneous pressure drop, and strong shear force [19,20,21]. The cavitation jet has the advantages of no exogenous chemicals, low processing temperature, and a small amount of nutrient loss [22,23]. The cavitation jet causes a strong destructive effect to reduce the size of large particles in the fluid so as to improve the functional properties of proteins. Cavitation jet technology is widely used in other industries, but it is rarely used in food processing applications. Wu et al. [24] used the cavitation jet to treat okara dietary fiber, which destroyed the structure of the dietary fiber, improving its water solubility index, water retention capacity, oil retention capacity, swelling capacity, and thickening capacity. Gong et al. [25] found that hydrodynamic cavitation and ultrasonic cavitation treatment unfolded the protein structure and exposed the hydrophobic residues, which affected the functional properties of proteins, so that the solubility, emulsifying properties, and foaming ability were significantly improved. Previous studies rarely investigated the relationship between the structural and emulsifying characteristics of oxidized SPI. Hence, a theoretical basis and guidance were required to better use SPI in food industry processes and storage.

Therefore, this study was performed to investigate the effect of cavitation jet treatment on the structure and properties of SPI oxidized by AAPH, thus finding ways to improve the performance of SPI by combining appropriate oxidation treatment and cavitation jet treatment, and also provide a theoretical basis for the development and storage of soy protein products.

## 2. Materials and Methods

### 2.1. Materials

SPI (purity 91.12%, 5.7% ash, 0.95% fat, and 0.87% crude fiber) was provided by Shandong Yuwang Ecological Food Industrial Co. (Yucheng, China), and AAPH was purchased from Sigma–Aldrich (Saint Louis, MO, USA). Soy oil was purchased from a local supermarket (Jiusan Cereals and Oils Industry Group Co., Ltd., Harbin, China). All other chemicals used were of analytical grade and deionized (DI) water was used throughout the experiment.

### 2.2. Protein Oxidation

The oxidative modification of SPI was performed as described by Wu et al. [14]. Soy protein solution (10 mg/mL suspended in 0.01 mol/L sodium phosphate buffer, containing 0.5 mg/mL sodium azide, pH 7.4) was mixed with serial concentrations of AAPH and then incubated in the air at 37 °C with continuous shaking in the dark for 24 h. The final concentration of AAPH was 0 (control), 0.2, 1, and 5 mmol/L. The reaction was stopped by immediately cooling the solution to 4 °C by ice-bathing and then centrifuging it at 8 kg for 15 min at 4 °C. The supernatant was dialyzed against DI water at 4 °C for 72 h to remove residual AAPH. The dialysis membrane had a molecular cut-off of 14,000 Da (Juyuang Biota Technology Co., Ltd., Shanghai, China).

### 2.3. Cavitation Jet Treatment of Oxidized SPI

A cavitation jet homogenizer (Beijing Zhongsen Huijia Technology Development Co., Ltd., Beijing, China) was used to homogenize the oxidized SPI solution. The processing pressure was 90 MPa, the processing time was 10 min, and the temperature was 25 °C. At the same time, protein solutions under different oxidation concentrations but not treated with the cavitation jet were taken as controls. Finally, the oxidized SPI solution that was treated by the cavitation jet and that was not treated by the cavitation jet was freeze-dried with a freeze dryer (ALPHA 1-4 LSC Freeze Dryer, Marin Christ Co., Ltd., Osterode, Germany) and stored at 4 °C until use. The natural SPI without oxidation and cavitation jet was taken as a control. The sample numbers for different oxidation and cavitation jet treatments are shown in Table 1.

### 2.4. Determination of Carbonyl Content

Protein carbonylation was quantified by the method described by Huang et al. [26]. First, 2,4-Dinitrophenylhydrazine (DNPT) was reacted with the carbonyl groups of oxidized SPI. The results were expressed as nmoles of carbonyl groups per milligram of soluble protein with a molar extinction coefficient of 22,000 M^−1^cm^−1^. The soluble protein concentration was evaluated by the biuret method with bovine serum albumin as the standard.

### 2.5. Characterization of Particle Size Distribution

The particle size distribution for SPI was determined by dynamic light scattering (DLS) with a particle size analyzer (Zetasizer Nano-ZS90, Malvern Instrument Co., Ltd., Worcestershire, UK) using the method proposed by Li Yang et al. [27]. SPI was suspended in a 0.01 mmol/L sodium phosphate buffer (pH 7.0) at a concentration of 0.2 mg/mL and stirred with a magnetic stirrer (S22-2 constant-temperature magnetic stirrer, Shanghai Sile Instrument Co., Ltd., Shanghai, China). Then, 0.8 mL of the protein solution was transferred to a square cuvette for DLS measurement. The results were expressed as volume percentage (%) versus particle size (nm).

### 2.6. Evaluation of Emulsifying Properties

#### 2.6.1. Preparation of Protein-Stabilized Emulsions

The emulsion preparation was conducted as described by Huang [28]. The emulsions were prepared from 10 mL of soy oil and 40 mL of various SPI dispersions (0.5%, *w/v*), adjusted to pH 7.0. To ensure that the emulsion system was fully mixed, a homogenizer (Ultra-Turrax T18 homogenizer, ANGNI Co. Ltd., Shanghai, China) was used for 2 min at 10,000 rpm. The final emulsions were used within 48 h.

#### 2.6.2. Determination of Mean Droplet Size and ζ-Potential of the Emulsion

A particle size analyzer (Zetasizer Nano-ZS90, Malvern Instrument Co., Ltd., Worcestershire, UK) was used to determine the droplet size distribution (single droplets or droplet aggregates) of a freshly prepared emulsion sample. The particle size was expressed as the volume-weighted mean diameter d_4,3_. This emulsion was diluted (50 μL in 5 mL of 0.01 mmol/L sodium phosphate buffer (pH 7.0)), and ζ-potential of the samples was determined using a zeta potential analyzer (Zetasizer Nano-ZS90, Malvern Instrument Co., Ltd., Worcestershire, UK).

#### 2.6.3. Emulsification Activity Index and Emulsification Stability Index

The emulsifying properties of SPI were determined with some modifications proposed by Pearce and Kinsella [29]. The freshly prepared homogeneous emulsion was taken out from the bottom and allowed to stand for 10 min. Aliquots (50 μL) of the emulsion were diluted 100 times with 0.1% (*w/v*) sodium dodecyl sulfate (SDS, pH 7.0). A microplate reader (Tecan Infinite M200 PRO microplate reader, Tecan Inc., Maennedorf, Switzerland) was used to measure the absorbance of a 0.1% SDS solution blank at 500 nm, and the emulsification activity index (EAI) and emulsification stability index (ESI) were calculated as follows:(1)$$\mathrm{EAI}=2\times 2.303\times \frac{A_{0}\times N}{C\times V\times 1000}$$
(2)$$\mathrm{ESI}=\frac{A_{0}\times ∆T}{∆A-A_{t}}=\frac{1}{A_{t}-A_{0}}\times ∆T$$
where *A*_0_ represents absorbance at time 0, *N* is the dilution factor (100), *C* is the protein concentration (g/mL) before emulsification (g/mL), *V* is the oil-phase volume fraction in emulsion (%), Δ*T* is the time difference (min), and Δ*A* represents the difference in absorbance within Δ*T*.

### 2.7. Determination of Sulfhydryl Groups (SH) and Disulfide Groups (S–S)

The contents of sulfhydryl groups (free and buried SH) and disulfide bond (S-S) in SPI were determined according to Huang et al. [26] using the 2,2′-dithiobis (5-nitropyridine) modified Ellman method, and the soluble protein concentration was evaluated by the biuret method with bovine serum albumin as the standard. According to the standard curve (y = 0.0502 × −0.0009, R2 = 0.9994), the protein concentration was calculated. The soluble protein contents were 12.75, 12.07, 15.13, 11.95, 15.67, 12.29, 16.09 mg/mL, respectively. The nanomoles of SH per milligram soluble protein were calculated using the extinction coefficient of 13,600 M^−1^ cm^−1^.

### 2.8. Measurement of Surface Hydrophobicity (H_0_)

Modifications were made by the method of Shen Lan et al. [30], and the *H*_0_ of the protein was determined using the fluorescent probe ANS^−^. Serial dilutions in 0.01 mmol/L phosphate buffer (pH 7.0) were prepared from SPI samples (stock solution; 1.5%, *w/v*) to a final concentration of 0.004–0.02% (*w/v*). A fresh ANS^−^ stock solution (0.08 mmol/L) was prepared using phosphate-buffered saline. An aliquot of 150 μL of the ANS^−^ stock solution was added to 15 μL of each diluted dispersion in a cell of a 96-well microplate and mixed. The fluorescence intensity was recorded at 370 nm (excitation) and 470 nm (emission) using a microplate reader (Tecan Infinite M200 microplate reader, Tecan Inc., Maennedorf, Switzerland). The initial slope of FI versus protein concentration (%, *w/v*) was calculated by a linear regression analysis and used as an index of *H*_0_.

### 2.9. Intrinsic Fluorescence Emission Spectroscopy

The intrinsic fluorescence was measured with a fluorescence spectrophotometer (F-2005, Hitachi Ltd., Tokyo, Japan). Protein solution (0.2 mg/mL) was prepared in 0.01 mmol/L phosphate buffer (pH 7.0). The excitation wavelength was 290 nm, the emission spectrum was recorded at the scanning speed of 10 nm/s in the range of 300–400 nm, and the excitation and emission were conducted under a constant 5-nm slit. Phosphate buffer was used as blank solution for all samples.

### 2.10. Circular Dichroism Spectra Measurement

The circular dichroism (CD) spectrum of SPI was measured with a Jasco J-815 Circular Dichroism Spectrophotometer (JASCO, Tokyo, Japan). Different SPIs were dispersed in 0.5 mg/mL deionized water. Quartz cuvettes were used at 20 °C with an optical path of 10 mm, and the CD spectrum was scanned at a wavelength in the far ultraviolet range (250–190 nm) and repeated three times at a speed of 100 nm/min and an interval of 0.5 nm. The content of secondary structure was calculated by the CONTINLL method using CDPro software (Narasimha Sreerama Research Group, Fort Collins, CO, USA).

### 2.11. Sodium Dodecyl Sulfate–Polyacrylamide Gel Electrophoresis

Sodium dodecyl sulfate–polyacrylamide gel electrophoresis (SDS-PAGE) analysis of proteins was performed as described by Laemmli [31]. SPI solution was suspended in phosphate buffer (0.02 mmol/L, pH 6.0) at a concentration of 1.5 mg/mL. SPI solution (1 mL) was added to 4 mL of sample buffer and heated for 5 min. The Bio-Rad MiniProtean3 system was used as a discontinuous buffer system on a 5% stacking gel and a 12% separating gel with a constant current set to 25 mA for approximately 1.5 h. The protein band was stained with Coomassie Brilliant Blue G-250 and then decolorized with 7% acetic acid (methanol: acetic acid: water, 227:37:236 (*v/v/v*)). A pre-stained protein molecular weight marker (10–170 kDa) was run on the same gel to identify the molecular weight of the protein bands in the stained gel.

### 2.12. Atomic Force Microscope

The microstructure of SPI samples was observed using an atomic force microscope (AFM). A protein suspension (5 µL) was dropped on the surface of the newly cut mica, dried in air at ambient temperature for 30 min, and used for AFM studies. AFM images were obtained using the BioScope (BS3-02) tapping mode of AFM (Veeco Instruments Inc., New York, CA, USA). The scanning area was 1 × 1 μm, and the scanning frequency was 1 Hz. The images were processed using NanoScope Analysis 1.7. For each preparation condition, the processing was repeated using at least two samples. Approximately 10 images were obtained for each formulation.

### 2.13. Statistical Analyses

All the results were recorded in at least triplicate. They were expressed as mean ± standard deviation. Means were compared using one-way analysis of variance followed by Duncan’s test (*p* < 0.05). Statistical analysis was done using SPSS (SPSS Inc., Chicago, IL, USA). The figures were made using Origin 9.5 (OriginLab Corporation, Northampton, MA, USA).

## 3. Results and Discussion

### 3.1. Effect of Cavitation Jet Treatment on the Carbonyl Content of Oxidized SPI

Protein carbonylation is an irreversible and non-enzymatic modification of proteins and is the most common indicator of protein oxidation [32,33]. As shown in Table 2, the carbonyl content increased significantly with the increase in AAPH concentration (*p* < 0.05). The results showed that peroxy radicals reacted with amino acid side chains of protein molecules under aerobic conditions, and some of them evolved into carbonyl derivatives in subsequent reactions, increasing the carbonyl content [34,35]. In contrast, the carbonyl content of oxidized proteins decreased significantly after cavitation jet treatment (90 MPa) (*p* < 0.05). Cavitation jet treatment produced strong shear force, high-speed impact force, and high-pressure transient release, leading to protein unfolding and carbonyl bond destruction [36]. However, the carbonylation degree of oxidized SPI treated with the cavitation jet was still higher compared with natural SPI, which might be due to the irreversible oxidative modification of oxidative stress [37].

### 3.2. Effect of Cavitation Jet Treatment on the Particle Size Distribution of Oxidized SPI

Protein particle size is an important factor affecting the functional properties of soy protein, and differences in particle size may lead to different functions [38]. The particle size distribution of the oxidized SPI after cavitation jet treatment is shown in Figure 1. The particle size of SPI increased significantly with the increase in the AAPH concentration (*p* < 0.05). The results suggested that oxidation could expand part of the protein structure, expose more hydrophobic groups, and increase the volume of the protein structure. AAPH induced the formation of soluble aggregates of soybean protein, and further oxidative expansion promoted the formation of insoluble precipitates and peroxidation of peptide chains [8,35].

After cavitation jet treatment, the particle size distribution of the oxidized protein changed. Both natural and oxidized SPI had a bimodal distribution with a wide particle size distribution. However, after the cavitation jet treatment, the particle size distribution tended to be unimodal. Meanwhile, Table 3 shows that d_4,3_ decreased sharply after cavitation jet treatment. This might be attributed to the strong shear and turbulent forces generated by microfluidization and cavitation. The hydrogen bond and hydrophobic interactions between SPI molecules were destroyed, which led to the decomposition of the aggregates formed by SPI oxidation into smaller particles [25].

### 3.3. Effect of Cavitation Jet Treatment on Hydrophobic Properties of Oxidized SPI

#### 3.3.1. SH and S-S Bond Contents

Cysteine residues are the most sensitive amino acid residues of proteins in the form of free SH groups or oxidized cysteine [14,39]. Table 2 shows free SH, total SH, and S-S group contents of untreated and cavitation jet-treated oxidized SPI samples. As the AAPH concentration increased, the content of the free SH group decreased significantly, and the content of the total SH group decreased significantly (*p* < 0.05). However, the content of S-S bonds gradually increased, but the differences in concentrations were not significant (*p* > 0.05). Sulfinyl radicals reacted with molecular oxygen to generate sulfhydryl peroxy groups, which further oxidized proteins. The SH groups were reduced and converted into reversible (protein disulfide and sulfenic acid) or irreversible forms (sulfinic and sulfonic acid) in different oxidation environments [40]. The results showed that not only the oxidation treatment led to the exchange between the SH groups and S-S bonds, but also some SH groups were converted into sulfur oxidation products. Cysteine residues and disulfide bonds are important in protein aggregation. The increase in the content of S-S bonds may cause changes in the SPI structure and lead to protein aggregation [41]. The content of free SH groups and/or S-S bonds of oxidized SPI were affected by the cavitation jet treatment. Compared with untreated SPI, the treatment significantly increased the free SH content but significantly decreased the content of the S-S bond (*p* < 0.05), but the difference between soybean proteins of different oxidation degrees was not significant (*p* > 0.05). The cavitation jet treatment did not significantly affect the total SH content, indicating that the reduction of free SH content at this time might be more inclined towards the spatial exposure of these groups rather than SH/S-S exchange. The cavitation jet unfolded the protein structure and exposed the SH groups, and the content of free sulfhydryl increased [25].

#### 3.3.2. Surface Hydrophobicity (*H*_0_)

Surface hydrophobicity is an important structural feature to evaluate the structural changes in proteins, which is closely related to the functional properties of proteins. As shown in Figure 2A, the *H*_0_ of oxidized SPI significantly increased. As the AAPH concentration increased, the *H*_0_ of oxidized SPI increased first and then decreased. *H*_0_ was affected by the balance between protein aggregation via hydrophobic interaction and partial protein unfolding. The increase in *H*_0_ might be related to the oxidative unfolding and/or denaturation of proteins, the decrease in the S-S bond content, and the exposure of hydrophobic domains initially embedded in protein molecules [8,39]. However, the unfolded or denatured proteins initially produced after oxidation treatment were thermodynamically unstable and associated with one another to form different aggregates through hydrophobic interactions between molecules. Therefore, when the AAPH concentration was 5 mmol/L, the decreased *H*_0_ might be due to the re-aggregation caused by covalent modification and the hydrophobic association of unfolded protein molecules [4]. Cavitation jet treatment (90 MPa) significantly increased the *H*_0_ of oxidized SPI (*p* < 0.05). The cavitation jet unfolded the oxidized protein structure, the particle size became smaller, the large aggregates in SPI changed into small aggregates, the intermolecular contact area increased, the hydrophobic amino acids were exposed, and *H*_0_ increased. In addition, the degree of improvement of *H*_0_ by the cavitation jet increased with the increase in AAPH concentration; that is, G (the concentration was 5 mmol/L) had the highest degree of improvement. Hence, the cavitation jet improved the surface hydrophobic properties of soybean protein with a higher aggregation degree more effectively. Shen et al. [30] studied SPI preheated using microfluidization treatment, which yielded similar results to the present study. As the temperature increased, the degree of protein aggregation increased, and the ability of the microjet to improve *H*_0_ increased.

#### 3.3.3. Intrinsic Fluorescence Emission Spectra

The maximum fluorescence emission wavelength (λ_max_) of tryptophan is related to the microenvironment of Trp, indicating that the relative position of the tryptophan residue in the protein is an indicator of the tertiary structural change in the protein [4,42]. It is therefore possible to determine the change in the tertiary structure of the protein from the fluctuations in fluorescence. As shown in Figure 2B, no significant difference was observed between oxidized SPI with 0.2 mmol/L AAPH and native SPI, indicating that a low concentration of peroxyl radical did not affect the tertiary structure of soy protein. When the concentration was 1 mmol/L, the fluorescence intensity increased significantly, but when the concentration was increased to 5 mmol/L, the fluorescence intensity decreased and was even lower than that of natural SPI. At the same time, the λmax of SPI was 335.8, 336.2, 336.7, and 332.5 nm, respectively, after treatment with 0, 2, 1, and 5 mmol/L AAPH. At first, when the AAPH concentration was relatively low, the oxidation induced protein unfolding, Trp was gradually exposed on the surface of the protein, *H*_0_ increased (Figure 2A), and the hydrophobic interaction between molecules increased, resulting in an increase in fluorescence intensity and a red shift of λ_max_. With a further increase in the degree of protein oxidation, some unfolded proteins or other hydrophobic groups are exposed to the hydrophilic environment, ultimately contributing to the formation of protein aggregates due to hydrophobic interactions. However, Trp was partially oxidized. Free radical destruction reduced the exposure of tryptophan residues originally located in polar environments, which were exposed to the nonpolar environment inside the molecule. Eventually, the λmax of the soy protein reduced and a blue shift occurred [4,35,43]. When the oxidative SPI was treated with the cavitation jet at a pressure of 90 MPa, the fluorescence intensity and λ_max_ significantly increased as predicted. On the contrary, cavitation jet treatment wholly increased the λ_max_ of oxidized SPI. This observation was consistent with the increased *H*_0_, indicating that cavitation jet treatment significantly increased the exposure of the hydrophobic domains to solvents in protein aggregates. These exposed hydrophobic domains tended to avoid contact with solvents. As a result, these small-sized new aggregates underwent structural rearrangement to make the structure more compact and bury more hydrophobic domains inside the molecule [22,44].

### 3.4. Effect of Cavitation Jet Treatment on the Emulsifying Properties of Oxidized SPI

#### 3.4.1. Emulsifying Capabilities

Emulsifying capability refers to the ability of an emulsifier to form and stabilize small droplets. Generally, the emulsifying capability is related to volume average particle diameter d_4,3_ and ζ-potential [30,35]. Therefore, the emulsifying capability of SPI was evaluated by measuring the d_4,3_ and ζ-potential of oxidized SPI emulsions treated with the cavitation jet (Table 3).

The d_4,3_ of the emulsion increased significantly with the increase in oxidant concentration (*p* < 0.05), and the trend of SPI dispersion was the same. The hydrophobic interaction was enhanced by oxidation treatment, leading to the aggregation of emulsified oil droplets and the increase in d_4,3_, which affected the emulsification performance. Moreover, d_4,3_ was further reduced by cavitation jet treatment. The formation of the protein structure induced by cavitation jet treatment exposed more hydrophobic groups, accelerated the movement of protein molecules, increased the interfacial tension of the emulsion, enabled the protein to diffuse at the oil–water interface better, and reduced the particle size.

The ζ-potential is another important indicator of the emulsifying ability, which indicates the surface charge property of the emulsion. As shown by d_4,3_ in Table 3, as the concentration of AAPH increased, the ζ-potential of the emulsion stabilized by oxidized SPI first increased and then decreased (*p* < 0.05). Chen et al. [35] speculated that amino acid residues greatly affected the surface charge and the exposure of previously buried amino acids and the ratio of acidic/basic amino acids caused by the selective oxidation of the susceptible amino acids resulted in the changes in the ζ-potential of the emulsion. This finding indicated that the surface hydrophobicity of the moderately oxidized protein increased, the electrostatic repulsion between the emulsion droplets, the surface charge was enhanced, and the emulsion stability was improved. The cavitation jet increased the surface charge of the moderately oxidized SPI emulsion and enhanced the electrostatic repulsion between the particles. This high electrostatic repulsion overcame the molecular interactions (such as van der Waals forces and hydrophobic interactions) which stabilized the existing protein aggregates. Therefore, it was possible to obtain greater emulsifying activity/stability. However, the effect of the cavitation jet on the over-oxidized emulsion was not significant.

#### 3.4.2. Emulsification Stability

EAI and ESI are always used to evaluate the emulsifying properties of proteins. EAI indicates the ability of the protein to form an oil–water interface, and ESI is the ability of the emulsion to stabilize at the oil–water interface after a predetermined period of time [45].

As shown in Figure 3, native SPI had EAI and ESI of 35.75 ± 2.78 m^2^/g and 45.78 ± 1.45%, respectively. The EAI and ESI of the emulsion stabilized by oxidized protein increased first and then decreased as the concentration of AAPH increased. This result indicated that moderate AAPH oxidation treatment effectively improved the EAI and ESI of SPI emulsion. This was because the low concentration of peroxy radical oxidation caused partial unfolding of the soy protein, the hydrophobic group inside the soy protein molecule was exposed, and the molecular flexibility was improved. Protein molecules were more easily combined with the oil–water interface, which increased the emulsification and emulsion stability of the oxidized soybean protein. The oxidation with high concentrations of peroxy radicals caused soy protein to form insoluble aggregates. Moreover, the surface hydrophobicity of oxidized soy protein decreased, leading to decreased flexibility and reduced surface area of oxidized soy protein molecules, thus resulting in a gradual decrease in emulsifying properties and emulsion stability [33]. When the oxidized SPI was treated with the cavitation jet at a pressure of 90 MPa, the cavitation jet increased the EAI of the emulsion (Figure 3A). At the same time, the degree of EAI increase was negatively related to the degree of oxidation. This result was contrary to *H*_0_’s result, indicating that, besides *H*_0_, other factors, such as the flexibility of SPI, also affected the emulsification performance. The flexibility of the protein was affected by noncovalent bonds such as hydrogen bonds, van der Waals forces, electrostatic attraction, and hydrophobic interactions. The cavitation and mechanical effects of cavitation jet processing destroyed these bonds, causing stretching of the tight structure of SPI. The flexible protein was quickly adsorbed to the interface, making it easier for SPI to form an interface film, thereby improving the emulsifying properties of SPI [38,46]. As shown in Figure 3B, the change trend of ESI was not consistent with that of EAI under cavitation jet processing. When the AAPH concentration was low, ESI significantly increased (*p* < 0.05). Ren et al. [25] studied the effects of hydrodynamic and ultrasonic cavitation on the emulsification properties of SPI. They found that as the treatment time increased, the soy protein particle size decreased and the solubility and *H*_0_ increased, leading to the formation of a β-sheet in the secondary structure. Swirling cavitation improved the emulsification properties of SPI. When the AAPH concentration was 5 mmol/L, the ESI decreased and became 50.97% ± 1.78%. It suggested that the cavitation jet processed the over-oxidized SPI, causing the oil droplets of the protein coating to flocculate. Moreover, the particle size became larger, the *H*_0_ decreased, the protein flexibility decreased, and the ESI deteriorated.

### 3.5. Effect of Cavitation Jet Treatment on the CD Spectrum of Oxidized SPI

CD spectroscopy can be used to reflect the changes in protein secondary structure. The secondary structure of oxidized SPI is shown in Table 4. The secondary structure of natural soy protein is composed of four different conformations, including α-helix, β-turn, β-sheet, and random structure. Compared with natural SPI, the oxidation of peroxyl free radicals reduced the ratio of α-helix and β-turn structures and increased β-sheet and random coil structures. The oxidation reaction caused protein unfolding, and the peptide bond was broken. When the exposed hydrophobic group increased to a certain value, the balance of electrostatic repulsion and hydrophobic interaction was broken, promoting protein aggregation and changes in the secondary structure [14,47]. After cavitation jet treatment, the β-sheet structure was further converted into β-turn and α-helical structure, the content of β-sheet and random coil structure decreased, and the content of β-turn and α-helix increased. The effect of the cavitation jet weakened the intermolecular interactions and exposed the nonpolar groups, leading to the weakening of the intermolecular hydrophobic interactions and breaking the intermolecular hydrogen bonds, resulting in an increase in α-helix and β-sheet structures. On the contrary, the cavitation jet treatment increased the ordered structure between protein molecules and protein aggregation, and hence the increase in α-helix and β-turn structures improved the functional properties of proteins [48,49]. At the same time, the higher the AAPH concentration, the greater the degree of change in the secondary structure caused by the cavitation jet. Hence, the cavitation jet more effectively improved the characteristics of soybean protein with a higher degree of aggregation.

### 3.6. Effect of Cavitation Jet Treatment on SDS-PAGE of Oxidized SPI

SDS-PAGE was used to further determine the oxidative aggregation behavior of soluble soy protein and the composition of subunits after cavitation jet treatment (Figure 4). The SDS-PAGE profile of native soy protein showed the characteristic bands of the subunits of β-conglycinin (α′, α, and β) and glycinin (As and Bs) (Figure 4, lane 7). After SPI was oxidized, as the AAPH concentration increased from 0.2 to 5 mmol/L, the intensity of α′, α, and As subunits of oxidized SPI by AAPH gradually decreased with the oxidation concentration. Concomitantly, some polymers (stacked at the top of the running gel) with high molecular weight were generated. β-Mercaptoethanol was added to the sample to break the disulfide bonds in the soybean protein before loading the protein sample. Therefore, the bands at the top of the gel were attributed to the aggregation of subunits connected by non-disulfide bonds. Chao et al. [50] believed that carbon–carbon covalent cross-linking was the predominant covalent cross-linking in oxidized proteins by peroxy radicals. More interestingly, the contents of β and Bs subunits increased first and then decreased (Figure 4, lane 1, 3, 5). The decreased contents of β and Bs subunits were related to protein fragmentation [51]. On the contrary, untreated and cavitated jet-treated SPI samples had similar typical SDS-PAGE profiles, and the cavitation jet did not significantly affect the molecular weight of oxidized SPI. This finding suggested that the effect of cavitation jet processing on S-S/SH interchange was dependent on the nature of the applied proteins in SPI.

### 3.7. Effect of Cavitation Jet Treatment on the Surface Morphology of Oxidized SPI

AFM is a powerful tool that can observe the surface morphology of proteins in an ambient atmosphere and can indirectly observe the dispersed and aggregated forms of proteins [52]. As shown in Figure 5, the 2D and 3D images of the oxidized SPI dispersion were obtained with NanoScope Analysis software. The natural SPI (A) was around 40–60 nm in height, most of which was in the form of spherical clusters, but the degree of clusters was relatively low, with some uneven surfaces. The particle morphology of the oxidized SPI dispersion became more uneven, and the particle size significantly increased. With the increase in AAPH concentration, the degree of particle clustering gradually increased, and most of the particles collided and aggregated to from larger irregular particles. The observations were in good agreement with the particle size distribution data (Figure 1). The larger oxidized SPI aggregates treated with the cavitation jet collapsed to form smaller particles and transformed to more uniform needle-like protrusions. The height distribution map corresponded to the particle size distribution map, and the distribution became narrower. This finding was similar to the results of Huang et al. [53] using the mechanical effect of ultrasonic cavitation to reduce the aggregation of protein SPI particles. This could be explained as the effects of strong shear force, high-speed impact, and high-frequency vibration generated during cavitation microjet processing, which could effectively reduce the particle size of proteins, destroy existing protein aggregation, and inhibit further aggregation of proteins.

## 4. Conclusions

In summary, our results show that the combination of cavitation jet and oxidation treatment had a great influence on the structure and emulsification properties of SPI. The cavitation jet reduced the size of protein aggregates due to the hydrodynamic shear forces associated with cavitation. The cavitation jet treatment destroyed the hydrophobic domain and greatly increased *H*_0_, resulting in an increase in the fluorescence intensity of SPI and a change in the tertiary structure. Of the treatment conditions tested, treatment with 1 mmol/L AAPH and 90 MPa cavitation jet produced smaller particle size, higher *H*_0_, and better emulsification properties (higher EAI and ESI). Overall, these results may help us to understand the effects of oxidation and cavitation jet treatment on the physicochemical and emulsification properties of SPI and provide ideas for improving the adverse effects of oxidation on SPI in industrial production.

## Figures and Tables

**Figure 1 foods-10-00002-f001:**
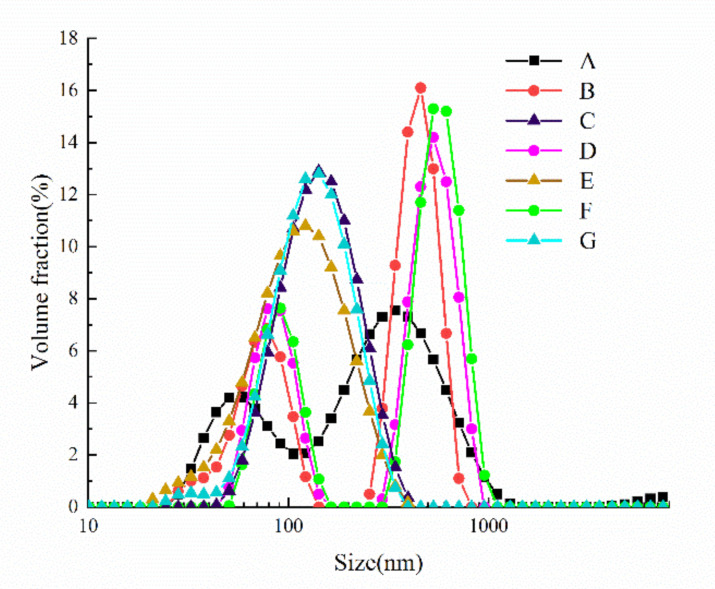
Particle size distribution of oxidized SPI before and after cavitation jet treatment.

**Figure 2 foods-10-00002-f002:**
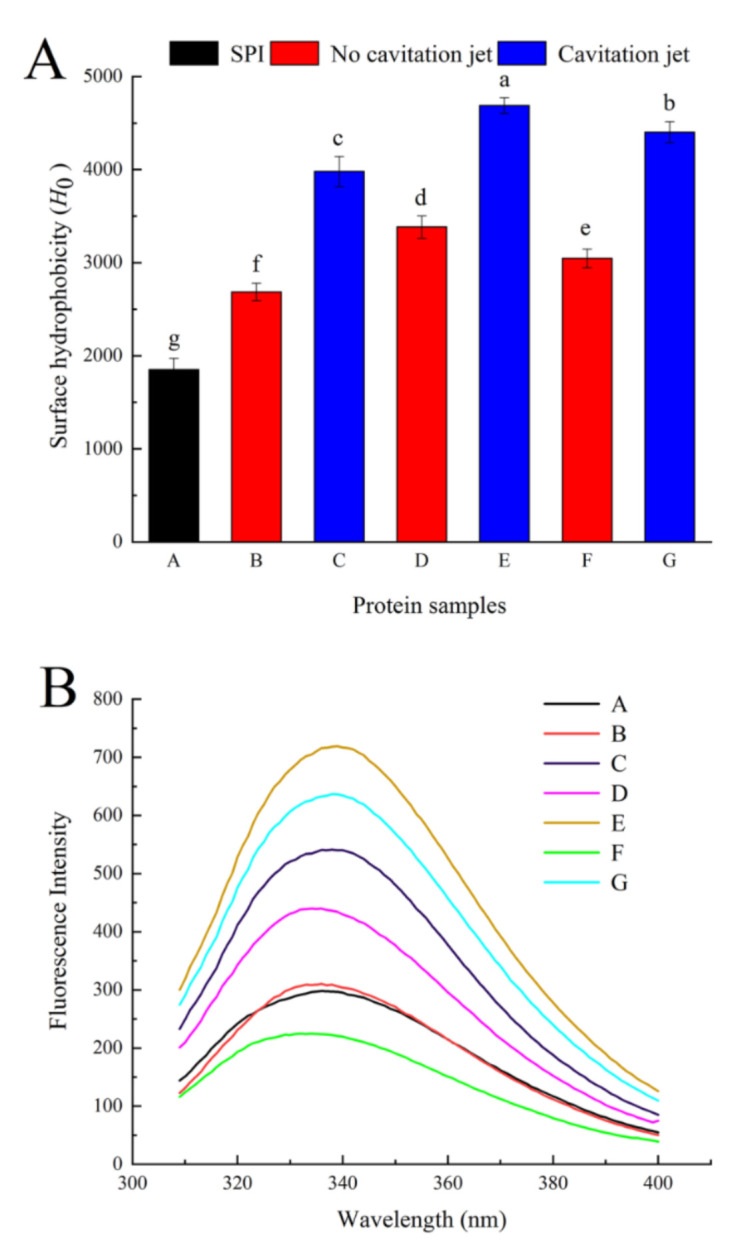
Surface hydrophobicity (*H*_0_, **A**) and intrinsic fluorescence emission spectrum (**B**) of oxidized SPI before and after cavitation jet treatment. Columns with different letters (a–g) are significant differences (*p* < 0.05).

**Figure 3 foods-10-00002-f003:**
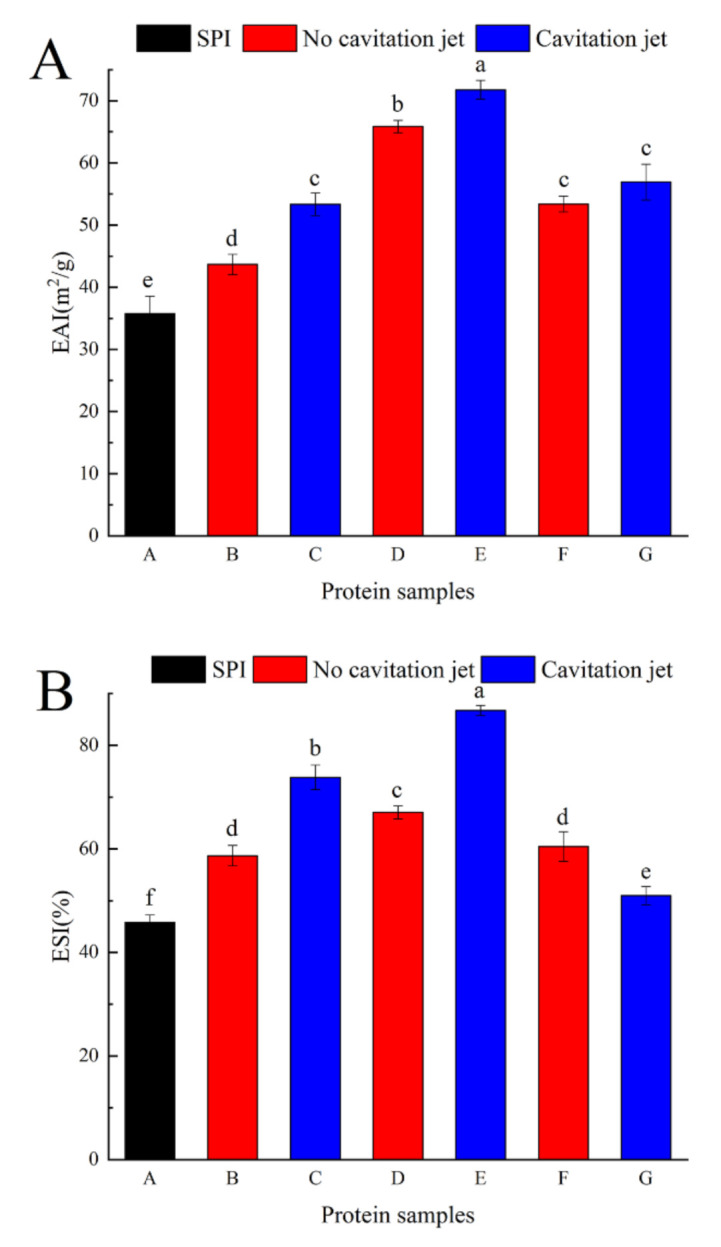
Emulsification activity index (EAI; **A**) and emulsion stability index (ESI; **B**) of SPI emulsions oxidized before and after cavitation jet treatment. Columns with different letters (a–f) are significant differences (*p* < 0.05).

**Figure 4 foods-10-00002-f004:**
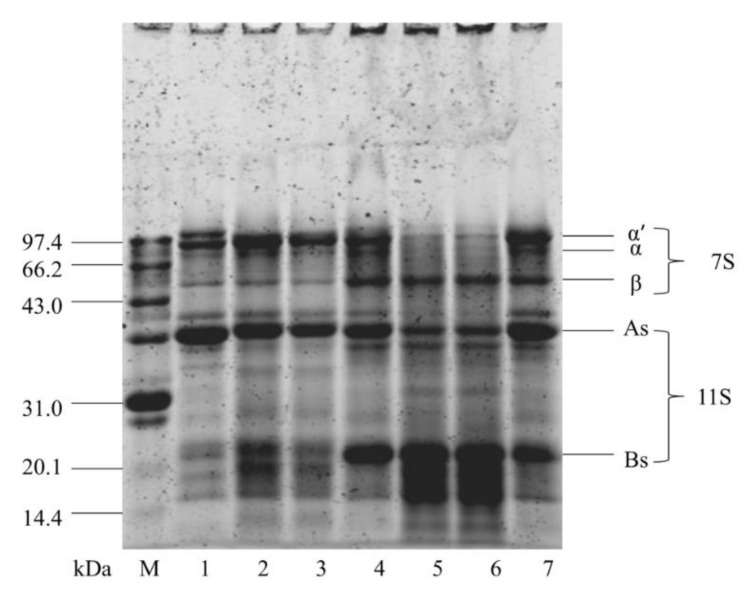
SDS-PAGE patterns of oxidized SPI before and after cavitation jet treatment. Band 1–7: B, C, D, E, F, G, and A. M: markers.

**Figure 5 foods-10-00002-f005:**
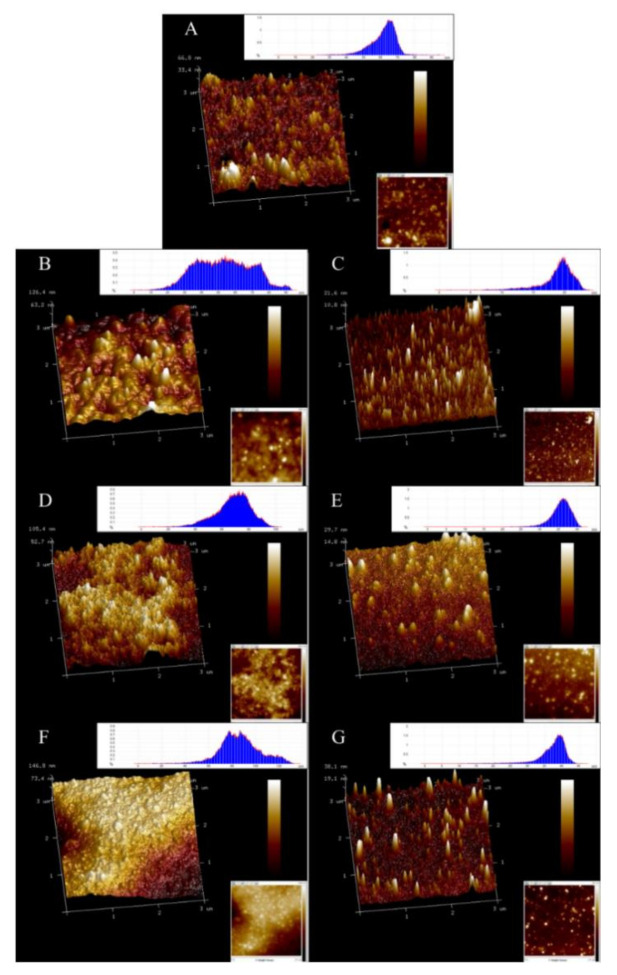
The 3D-view height-view (bottom right illustration) and cross-section (upper right illustration) Atomic Force Microscope (AFM) image of SPI oxidized before and after cavitation jet treatment. Note: Scale bar is 3 μm. (**A**)–(**G**) represents protein samples.

**Table 1 foods-10-00002-t001:** Different oxidation and cavitation jet treatment sample numbers.

Parameter	Sample Serial Number						
	A	B	C	D	E	F	G
AAPH concentration (mmol/L)	0	0.2	0.2	1	1	5	5
Cavitation jet pressure (MPa)	0	0	90	0	90	0	90

AAPH: 2,2′-azobis (2-amiylpropane) dihydrochloride.

**Table 2 foods-10-00002-t002:** The carbonyl, free sulfhydryl (SH), total SH groups, and S-S bond content of oxidized soy protein isolate (SPI) before and after cavitation jet treatment.

Protein Samples	Carbonyl (nmol/mg)	Free SH (μmol/mg)	Total SH (μmol/mg)	S-S (μmol/mg)
A	5.67 ± 0.03 ^f^	7.07 ± 0.03 ^a^	9.73 ± 0.03 ^a^	3.80 ± 0.02 ^b^
B	5.86 ± 0.02 ^c^	6.39 ± 0.03 ^c^	9.70 ± 0.03 ^ab^	3.89 ± 0.03 ^ab^
C	5.77 ± 0.02 ^d^	6.89 ± 0.01 ^b^	9.71 ± 0.04 ^a^	3.83 ± 0.02 ^ab^
D	5.95 ± 0.03 ^b^	5.88 ± 0.02 ^f^	9.67 ± 0.02 ^ab^	3.93 ± 0.03 ^ab^
E	5.71 ± 0.02 ^e^	6.01 ± 0.02 ^d^	9.69 ± 0.01 ^ab^	3.88 ± 0.03 ^ab^
F	6.03 ± 0.01 ^a^	5.22 ± 0.02 ^g^	9.64 ± 0.03 ^b^	3.97 ± 0.03 ^a^
G	5.72 ± 0.01 ^e^	5.95 ± 0.02 ^e^	9.67 ± 0.05 ^ab^	3.90 ± 0.03 ^ab^

Values in the same column followed by different letters (^a^–^g^) are significant difference (*p* < 0.05).

**Table 3 foods-10-00002-t003:** The d_4,3_ of the oxidized SPI dispersion and the d_4,3_ and ζ-potential of the emulsion droplets before and after the cavitation jet treatment.

Protein Samples	Dispersion	Emulsion Droplet	
	d_4,3_ (nm)	d_4,3_ (μm)	ζ-Potential (mV)
A	266.21 ± 7.94 ^f^	4.32 ± 0.04 ^f^	−22.53 ± 0.81 ^b^
B	355.03 ± 9.16 ^d^	4.69 ± 0.08 ^e^	−23.87 ± 1.03b ^c^
C	306.09 ± 3.95 ^e^	4.14 ± 0.08 ^g^	−23.97 ± 1.21b ^c^
D	439.38 ± 7.59 ^b^	5.62 ± 0.15 ^c^	−25.03 ± 0.58 ^c^
E	389.05 ± 4.52 ^c^	5.02 ± 0.08 ^d^	−27.69 ± 0.98 ^d^
F	503.84 ± 5.73 ^a^	6.59 ± 0.09 ^a^	−20.80 ± 0.61 ^a^
G	434.26 ± 4.58 ^b^	5.87 ± 0.08 ^b^	−20.73 ± 1.12 ^a^

Values in the same column followed by different letters (^a^x2013;^g^) are significant difference (*p* < 0.05).

**Table 4 foods-10-00002-t004:** The secondary structure content of oxidized SPI before and after cavitation jet treatment.

Protein Samples	α-Helix (%)	β-Sheet (%)	β-Turn (%)	Random Coil (%)
A	19.07 ± 0.04 ^b^	22.82 ± 0.09 ^c^	16.04 ± 0.08 ^d^	42.93 ± 0.09 ^f^
B	17.20 ± 0.11 ^c^	21.81 ± 0.06 ^d^	16.79 ± 0.09 ^c^	44.17 ± 0.07 ^e^
C	19.58 ± 0.06 ^a^	20.12 ± 0.09 ^f^	17.90 ± 0.10 ^b^	42.35 ± 0.06 ^g^
D	13.78 ± 0.06 ^e^	23.63 ± 0.07 ^a^	12.58 ± 0.07 ^g^	50.03 ± 0.11 ^a^
E	17.27 ± 0.07 ^c^	22.97 ± 0.07 ^b^	14.46 ± 0.08 ^f^	45.25 ± 0.05 ^d^
F	16.14 ± 0.10 ^d^	21.15 ± 0.05 ^e^	15.07 ± 0.09 ^e^	48.66 ± 0.08 ^b^
G	19.71 ± 0.09 ^a^	15.66 ± 0.07 ^g^	18.66 ± 0.08 ^a^	45.95 ± 0.05 ^c^

Values in the same column followed by different letters (^a^–^g^) are significant differences (*p* < 0.05).

## Data Availability

No new data were created or analyzed in this study. Data sharing is not applicable to this article.

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
