# Peer review of "Effects of Cavitation Jet Treatment on the Structure and Emulsification Properties of Oxidized Soy Protein Isolate"

_foods, 2020, doi:10.3390/foods10010002_

Round 1
Reviewer 1 Report
The foods-1009389 manuscript I can say it is a very good one. The experimental design is suitable, the authors used several and proper methods, and they used the appropriate statistical analysis. Moreover, they explained the results and came to the correct conclusions.
Author Response
Dear Reviewer:
Thank you for your comment on our manuscript entitled " Effects of cavitation jet treatment on the structure and emulsification properties of oxidized soy protein isolate" (No. foods-1009389). Those comments are all valuable and very helpful for revising and improving our paper, as well as the important guiding significance to our researches. According to the comments, we have carefully modified the manuscript and made correction, which we hope meet with approval.
Yours sincerely,
Fei Teng
E-mail: ([email protected])

Reviewer 2 Report
This study uses a wide range of physical and other measurements to investigate the effect of jet cavitation on both oxidised and non-oxidised SPI. The standard of experimentation is high and I have no comments on this.
There is however too much focus on describing the results and not enough on explanation and thus does not help realise the full potential of the paper. This is probably best summarised by the phrase in the Conclusions : " these results may help understand the effect of cavitation jet treatment on SPI modification ". It is not unreasonable to suggest that this was in fact the purpose of this study. Although, I suggest a major revision, it should not be onerous to provide a more in-depth treatment in the Conclusions which gives a more mechanistic explanation.
My other major comment concerns Table 2 which shows the effect of increasing AAPH oxidation concentration and jet power on markers of oxidation. My concern, for example, is that despite a 25 fold increase in AAPH , the carbonyl shows only a 5% increase. It appears that it is not possible to change the level of oxidation significantly and yet this is intended to be a major aim of the study.
Author Response
Dear Reviewer:
Thank you for your comment on our manuscript entitled " Effects of cavitation jet treatment on the structure and emulsification properties of oxidized soy protein isolate" (No. foods-1009389). Those comments are all valuable and very helpful for revising and improving our paper, as well as the important guiding significance to our researches. According to the comments, we have carefully modified the manuscript and made correction, which we hope meet with approval. Revised portions are marked in red in the paper. The main corrections in the paper and the responds to the comments are as following:
Point 1: There is however too much focus on describing the results and not enough on explanation and thus does not help realise the full potential of the paper. This is probably best summarised by the phrase in the Conclusions : " these results may help understand the effect of cavitation jet treatment on SPI modification ". It is not unreasonable to suggest that this was in fact the purpose of this study. Although, I suggest a major revision, it should not be onerous to provide a more in-depth treatment in the Conclusions which gives a more mechanistic explanation.
Response 1: Thank you for your valuable comments. Based on your comments, the conclusions have been revised. The conclusion part now reads: " In summary, our results show that the combination of cavitation jet and oxidation treatment had a great influence on the structure and emulsification properties of SPI. The cavitation jet reduced the size of protein aggregates due to the hydrodynamic shear forces associated with cavitation. The cavitation jet treatment destroyed the hydrophobic domain and greatly increased H0, resulting in an increase in the fluorescence intensity of SPI and a change in the tertiary structure. Of the treatment conditions tested, treatment with 1 mmol/L AAPH and 90 MPa cavitation jet produced smaller particle size, higher H0 and better emulsification properties (higher EAI and ESI). Overall, these results may help to understand the effects of oxidation and cavitation jet treatment on the physicochemical and emulsification properties of SPI, and provide ideas for improving the adverse effects of oxidation on SPI in industrial production."(L448-450)
Point 2: My other major comment concerns Table 2 which shows the effect of increasing AAPH oxidation concentration and jet power on markers of oxidation. My concern, for example, is that despite a 25 fold increase in AAPH , the carbonyl shows only a 5% increase. It appears that it is not possible to change the level of oxidation significantly and yet this is intended to be a major aim of the study.
Response 2: Thank you very much for your careful review. Due to our negligence, we did not consider this issue. We have checked the relevant literature, and we have similar results[1-3]. For example, Chen et al.[1] found that untreated protein has no significant increase in carbonyl compared with low-oxidation concentration conditions. When the AAPH concentration was 5 mmol/L, the carbonyl content only increased by 0.26 nmol/mg compared with the untreated sample. Ye et al.[2] used AAPH to oxidize peanut protein isolate, and the increase in carbonyl content was relatively small. Although the increase in this experiment was small, it was still significantly larger than that of untreated. At the same time, subsequent studies also showed that oxidation treatment had a certain effect on the structure and emulsification properties of soybean protein. In addition, part of the reason may be that our unprocessed protein was placed in the air during the preparation process and subsequent determinations, resulting in slight oxidation. We will pay attention to this problem in future experiments.
- Chen, N.; Zhao, M.; Sun, W.; Ren, J.; Cui, C., Effect of oxidation on the emulsifying properties of soy protein isolate. Food Research International 2013, 52 (1), 26-32.
- Ye, L.; Liao, Y.; Zhao, M.; Sun, W., Effect of Protein Oxidation on the Conformational Properties of Peanut Protein Isolate. Journal of Chemistry 2013, 2013, 1-6.
- Wu, W.; Hou, L.; Zhang, C.; Kong, X.; Hua, Y., Structural modification of soy protein by 13-hydroperoxyoctadecadienoic acid. European Food Research and Technology 2009, 229 (5), 771-778.

Reviewer 3 Report
The paper describes a very interesting and important topic dealing with the use of innovative technologies to improve quality and functional properties of soy protein isolate. Generally, the paper is very well written and addresses all the important aspects of the study. I have the following comments:
Introduction, L32-34: too long sentence, I suggest to split it into 2 sentences.
L104-105: more details needed for the method description. The sample preparation prior to the analysis is missing (which may be different from the one mentioned in the reference).
L116: The abbreviation DLS should be explained in the L111 when you first time mention it in the text.
L 149: you mean BioRad method? Please provide the reference for soluble protein concentration.
L240: increased, but not mentioned significantly or not. Please correct it.
L249-250: are you sure it was significant increase? What analysis you applied to check the significance?
Author Response
Dear Reviewer:
Thank you for your comment on our manuscript entitled " Effects of cavitation jet treatment on the structure and emulsification properties of oxidized soy protein isolate" (No. foods-1009389). Those comments are all valuable and very helpful for revising and improving our paper, as well as the important guiding significance to our researches. According to the comments, we have carefully modified the manuscript and made correction, which we hope meet with approval. Revised portions are marked in red in the paper. Please refer to the attachment for major corrections and responses to comments in this article. Thank you.
Yours sincerely,
Fei Teng
E-mail: ([email protected])

Round 2
Reviewer 3 Report
I think the paper can now be accepted for publication.